# Influence of Aggregate Pollution in Truck Escape Ramps on Stopping Distance of Uncontrolled Vehicles

**Pinpin Qin \*, Ziming Li, Hao Li, Junming Huang and Guiqi Wang**

School of Mechanical Engineering, Guangxi University, Nanning 530004, China
\* Correspondence: qpinpin@gxu.edu.cn

**Abstract:** Migration of fine materials such as soil from the roadbed and the ground will gradually pollute the aggregate in the arrester bed of truck escape ramps. However, there are few studies on the impact of aggregate pollution of the arrester bed on the stopping distance of runaway vehicles. This paper uses the discrete element method to study the influence of aggregates with different degrees of pollution on stopping distance by taking silty cohesive soil as a typical pollutant. In this paper, the stopping process of the uncontrolled vehicle on the arrester bed with different pollution levels was numerically simulated. The simulation results show that the uncontrolled vehicle's stopping distance increases with the contaminated aggregate's soil content. The simulation results show that when the soil content in the contaminated aggregate is less than 15%, the increase in the stopping distance of the uncontrol vehicle is less than 5%; when the soil content is 20–25%, the stopping distance of the uncontrolled vehicle increases by more than 20%; and when the soil content is 30–35%, the stopping distance of uncontrol vehicle increases by more than 50%. Different maintenance measures should be taken according to the increase in stopping distance.

**Keywords:** truck escape ramps; discrete element method; numerical simulation; aggregate pollution; stopping distance





## 1. Introduction

Road traffic safety is one of the basic requirements to achieve sustainable development. Road safety infrastructure is of great significance in road traffic safety. Various countries and regional organizations have emphasized the importance of planning and designing road safety infrastructure and its role in achieving sustainable development [1–3]. Decades of engineering practice experience at home and abroad has proved that the refuge lane is the most effective road safety infrastructure to curb serious traffic accidents caused by uncontrolled vehicles for long downhill sections. Since the invention of truck escape ramps in 1956, there has been much research on the location, laying length, slope, and aggregate selection of the truck escape ramps. Al-Qadi sampled the aggregate of the arrester bed. The microscopic parameters of the aggregate particles were measured by a dynamic triaxial compression test and static triaxial compression test in the laboratory. A two-dimensional 'wheel-aggregate' model was established according to the interaction between the wheel and the aggregate. Al-Qadi verified the accuracy of the 'wheel-aggregate' model through the tests [4,5]. With the development of computer science, numerical simulation has become a common method to study various engineering problems. Computer numerical simulation can significantly reduce the high investment and risk operation of actual vehicle tests. Inspired by molecular dynamics, Cundall proposed the discrete element method. The discrete element method has since been widely used in the study of tire–soil mechanics [6–8] and truck escape ramps [9,10]. Nishiyama et al. combined the finite element method with the discrete element method [11–13], using joint simulation to study the wheel–soil problem. In the simulation process, Nishiyama only activated the discrete element particles in a small range near the wheel, so that the activation area changed with the position of the wheel.

This method greatly reduces the amount of calculation. Based on the discrete element method, Liu Pan simulated the stopping process of an uncontrolled vehicle on the truck escape ramps through numerical simulation, and verified the numerical simulation with a real vehicle test. The test results verified the feasibility of using the discrete element method to simulate the related problems of truck escape ramps [14]. In the method of arrester bed length proposed by the U.S. Federal Highway Administration, the rolling resistance was set to a constant value, and the calculation error was significant without considering the wheel subsidence and the effect of discrete aggregates. Greto K proposed a calculation method based on the FOSM method, which introduced the influence of aggregate random variables. The calculation is more complex, but the calculation accuracy is higher [15]. Qin et al. [16] reduced the amount of particle calculation by improving the aggregate generation method and established a three-dimensional model of the stopping process of uncontrol vehicle in the truck escape ramps. They studied the influence of the sinkage of different vehicle weights and driving speeds on the stopping distance, and gave suggestions on the laying length and thickness of the brake bed. Miha Ambrož used a passenger car to carry out a real vehicle test of emergency stop in truck escape ramps. According to the test results, they concluded that: when the roundness of the aggregate particles of the arrester bed is large and the particle size is small, the deceleration after the vehicle enters is greater [17]. Filippo A et al. made deformable units through cementitious materials to replace the gravel aggregate of the traditional truck escape ramps. The experimental results show that the new arrester bed made of the material can stop the uncontrolled vehicle with 40 tons of weight and 56 km/h speed. The new arrester bed had a good performance, but its high cost had not been widely promoted [18]. In summary, the current research on truck escape ramps is based on the assumption that truck escape ramps are in an ideal state. In practical engineering applications, there are many problems in truck escape ramps, such as soil-contaminated aggregate, fuel and goods entering the vehicle leaking contaminated aggregate, and gravel aggregate corrosion degradation, resulting in the performance degradation of the truck escape ramps. The performance of the truck escape ramps will inevitably decline over time. So, the stopping distance of the uncontrolled vehicle on the escape ramp increases, resulting in accidents. There are few studies on the influence of the performance degradation of truck escape ramps on the stopping distance of uncontrolled vehicles. The truck escape ramp's maintenance depends on an engineer's experienced judgment. Based on the discrete element method, this paper studies the influence of performance degradation caused by soil pollution in truck escape ramps on stopping distance and provides the relevant theoretical basis for the maintenance of truck escape ramps.

## 2. Methodology

The discrete element method was proposed by American scholar Cundall. The discrete element method divides a discontinuous discrete medium into small particles. The particles interact with each other through a contact model. The fine-scale variation between particles constitutes the macroscopic physical properties of the medium. Then, Cundall and Hart gave a systematic and detailed description of the discrete element method. In 1992, Cundall's ItascaR company developed a series of particle discrete element software (PFC2D and PFC3D). The basic assumptions in numerical simulation of the discrete element method are as follows: (1) The sphere is the basic calculation unit. (2) The spherical element is a rigid body without deformation but will overlap in the calculation process, and the contact force is calculated by overlap. (3) The motion of the sphere unit conforms to Newton's law of motion. The discrete element method is widely used to simulate discontinuous media such as soil and rock.

### 2.1. Constitutive Model

Many constitutive models in the discrete element method can be used to simulate different materials. This paper selects the linear model and linear contact bond model

to construct the aggregate–wheel model. The structure of the linear model is shown in Figure 1. The normal force in the linear model can be described as:

$$F_n = \frac{k_{n,A}k_{n,B}}{k_{n,A} + k_{n,B}}x_n + \eta_n x_n \tag{1}$$

where $k_{n,A}$ and $k_{n,B}$ are the normal stiffnesses of particle A and the normal stiffnesses of particle B, respectively; $x_n$ is normal contact displacement; and $\eta_n$ represents the normal damping between particles. The tangential force in the linear model can be described as:

$$F_s \begin{cases} \Delta F_s = \frac{k_{s,A}k_{s,B}}{k_{s,A} + k_{s,B}}x_s + \eta_s x_s & |F_s| \leq \mu|F_n| \\ sign(x_s).\mu|F_n| & |F_s| \geq \mu|F_n| \end{cases} \tag{2}$$

where $k_{s,A} \neq k_{s,B}$ represent the tangential stiffness of particle A and the tangential stiffness of particle B respectively; $x_s$ is the tangential relative displacement; $\eta_s$ represents the tangential damping between particles; and $\mu$ is the friction coefficient between particles. When $|F_s| \leq \mu|F_n|$, the tangential force between particles is determined by stiffness and damping; when $|F_s| \geq \mu|F_n|$, the tangential force among particles is determined by the friction coefficient. The linear model and the physical properties of gravel are the same, and this paper uses the linear model to simulate gravel aggregate.

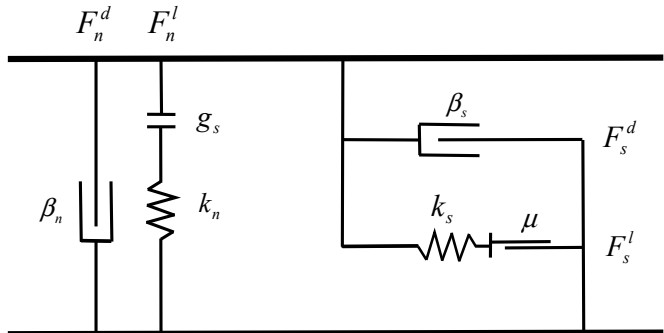

**Figure 1.** Linear Model.

The structure of the linear contact bonding model is shown in Figure 2. The linear contact bonding model has corresponding bond failure conditions in the normal and tangential directions. If the tensile strength limit is exceeded under tension, the bond is broken; if the bond is not broken in tension, the limit of shear strength should be strengthened. If the bond strength exceeds the tangential bond strength, the bond is broken during shear. After the bond is broken, the linear contact bond model becomes the linear model, and the constitutive characteristics of the linear contact bond model are shown in Figure 3. The linear contact bond model is consistent with the physical properties of cohesive soil, so the linear contact bond model is often used to simulate cohesive soil.

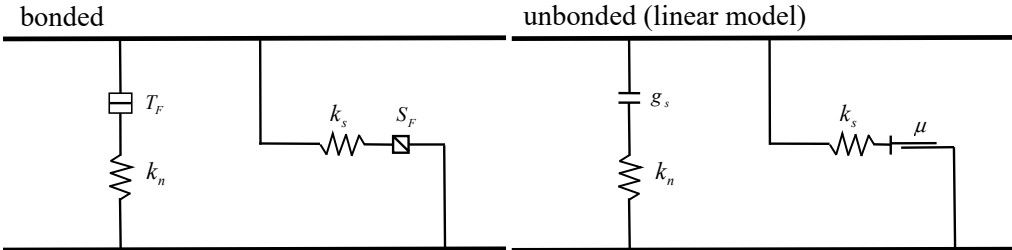

**Figure 2.** Linear Contact Bond Model.

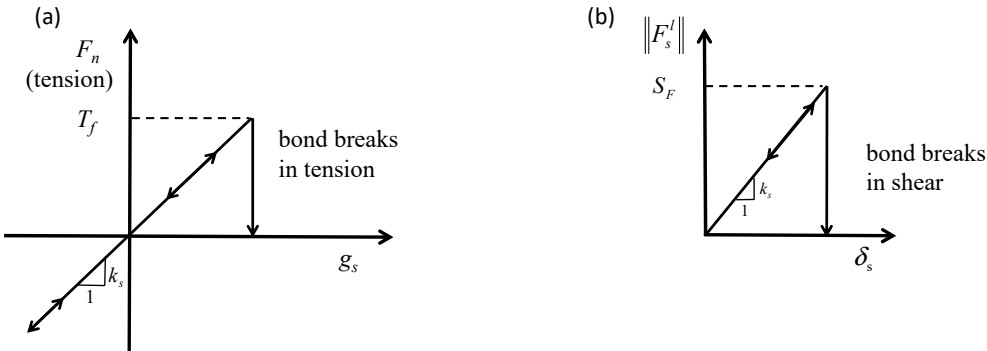

**Figure 3.** Force–displacement law for the linear component of the bonded linear contact bond model. (**a**) Normal force versus surface gap; (**b**) Shear force versus relative shear displacement.

### 2.2. Build Wheel–Aggregate Model

The number of particles in the full-size aggregate model is too large. In order to reduce the amount of calculation, this paper refers to the slave area algorithm in reference [19] to generate the whole segment of aggregate. The specific process is as follows: (1) According to the calculation capacity and efficiency of our computer, we choose to generate 5 m-long aggregates first. When the wheel reaches the midpoint of the aggregate, the program is suspended and the data information such as the position, speed, and contact of the model at this time is saved. (2) Then, generate the next equal length aggregate, and continue the procedure when the contact force of the next segment of aggregate particles is balanced. (3) Finally, after the wheels reach the next aggregate section, the program is suspended, the previous aggregate section is deleted, and the next aggregate section is generated simultaneously. Cycle this operation until the end of the simulation process.

### 2.3. Gravel Model Calibration

The research object of this paper is the gravel aggregate truck escape ramps which are most widely used. In order to ensure the accuracy of the numerical simulation, the microscopic parameters of gravel are calibrated by a numerical simulation biaxial compression test. Calibration test data are taken from the literature [4]. A biaxial compression test model, the same size as the laboratory biaxial compression, is established in PFC2D software, as shown in Figure 4. Three groups of numerical simulation tests are set up: (1) the relative density was 13%, and the confining pressure was 69.0 pa; (2) the relative density is 45%, and the confining pressure is 137.9 kpa; (3) the relative density is 80%, and the confining pressure is 206.9 kpa. The numerical simulation test conditions are consistent with the laboratory tests. See Table 1 for the numerical changes in peak stress, peak strain, and volume strain of the numerical simulation biaxial compression test and laboratory test data. It can be seen from the table that the test results are consistent, and the calibration of microscopic parameters of aggregate particles meets the requirements. See Table 2 for the calibration results of microscopic parameters of gravel particles. In the numerical simulation test, the peak stress of the test group with a relative density of 13% is slightly higher than that of the test group with a relative density of 45% because there is a critical point of friction and interlocking action between particles. When the pressure loaded in the tests exceeds the critical point, the peak pressure will change abruptly and then fall back.

### 2.4. Wheel Model Calibration

The wheel model uses a large disk particle for simulation. The calculation method of stopping distance is as follows: the starting point is the position at which the wheel is given the initial driving speed, and the ending point is the position at which the wheel finally stops after consuming energy on the arrester bed of the truck escape ramps. The driving distance of the wheel on the way is the stopping distance of the vehicle. In order to obtain reasonable parameters for the wheel model, the driving speed and the arrester bed slope of

the numerical simulation test and the actual vehicle test are set to be consistent, and the microscopic parameters of the wheel model are adjusted and calibrated by comparing the stopping distance and sinkage in the actual vehicle test. The calibration is completed when the data results of the numerical simulation stopping test and real vehicle test are similar. The actual vehicle test data are taken from reference [4]. The comparison results between the actual vehicle test and the numerical simulation are shown in Figure 5. The microscopic parameters of tire particles are shown in Table 3.

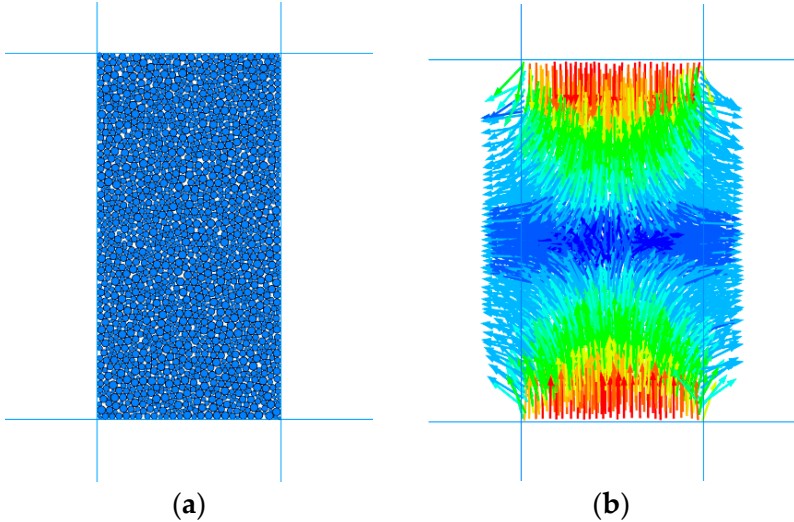

(a) (b)

**Figure 4.** (**a**) Biaxial compression test. (**b**) Displacement vector diagram of biaxial compression test.

**Table 1.** Calibration results of gravel aggregate particles.

| Relative Densities | Confining Pressures (kPa) | Peak Stress (kPa) | | Peak Strain (%) | | Volumetric Strain (%) | |
|---|---|---|---|---|---|---|---|
| | | Real Data | Simulation Data | Real Data | Simulation Data | Real Data | Simulation Data |
| 13% | 69.0 | 294.3 | 274.16 | 4.4 | 6.66 | −0.225 | 0.15 |
| | 137.9 | 480.7 | 512.25 | 5.5 | 6.05 | −0.290 | 0.197 |
| | 206.9 | 571.1 | 743.62 | 6.3 | 7.55 | −0.86 | 0.194 |
| 45% | 69.0 | 324.3 | 261.3 | 3.9 | 6.38 | 0.105 | 0.11 |
| | 137.9 | 659.3 | 488.87 | 5.5 | 6.56 | 0.13 | 0.19 |
| | 206.9 | 701.7 | 742.43 | 5.8 | 6.75 | 0.19 | 0.22 |
| 80% | 69.0 | 367.8 | 316.84 | 3.3 | 4.83 | 0.615 | 0.14 |
| | 137.9 | 772.4 | 668.77 | 4.1 | 5.12 | 0.65 | 0.23 |
| | 206.9 | 986.1 | 999.71 | 5.3 | 7.48 | 2.17 | 1.46 |

**Table 2.** Micro-parameters of gravel particles.

| Average Particle Size (cm) | Particle Density (kg/m$^3$) | Normal Stiffness (N/m) | Tangential Stiffness (N/m) | Friction Coefficient |
|---|---|---|---|---|
| 2 | 2630 | $3 \times 10^6$ | $2.45 \times 10^6$ | 30 |

The calibration results show that the stopping distance data of the numerical simulation test and the real vehicle test are basically consistent in a reasonable range. However, the calibration error of the stopping distance in the experiment is small, and the sinkage error is large. There are two main reasons: (1) In the real vehicle experiment, because the ruts are rolled by the front wheel and the rear wheel will be rolled repeatedly, the numerical simulation settlement value is generally smaller than that. (2) The numerical experiment adopts a two-dimensional model, which is different from the actual particles in terms of fluidity.

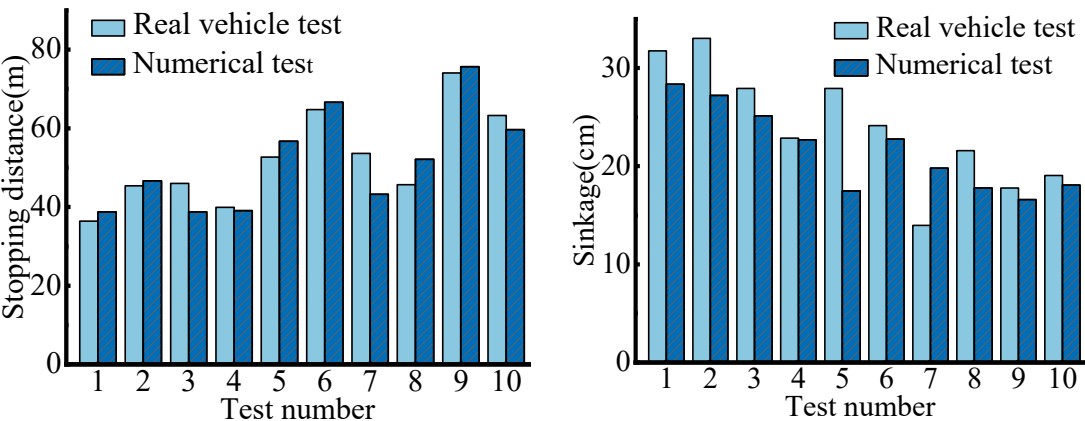

**Figure 5.** Calibration results of real vehicle test.

**Table 3.** Microparameters of wheel particles.

| Wheel Radius (m) | Normal Stiffness (N/m) | Tangential Stiffness (N/m) | Friction Coefficient |
|---|---|---|---|
| 0.55 | $6 \times 10^6$ | $1.39 \times 10^7$ | 0.5 |

*2.5. Calibration of Soil Pollutants*

In order to ensure the accuracy of microscopic parameters of the soil pollutants, the soil was calibrated using numerical simulation direct shear test, as shown in Figure 6. The soil calibration in this paper will focus on the cohesive soil widely distributed in Southwest China. After obtaining the soil sample, because the soil moisture content is uneven and difficult to measure, and the soil structure will be damaged during the sampling process, the direct shear test of the silty cohesive soil remolded in the laboratory is selected for soil calibration. The dry density of the direct shear test sample is 1.54 g/cm$^3$, the moisture content is 18%, and the consolidation ratio is 1. After standing for 24 h, the moisture content is uniform. The test data are from the measured test in reference [20]. The numerical simulation test conditions are consistent with the laboratory tests. The comparison of the shear stress displacement curve between the numerical simulation direct shear test and the laboratory direct shear test under vertical pressure of 60 kpa, 120 kpa, 180 kpa, and 240 kpa is shown in Figure 7. The shear stress–displacement curve and peak shear stress of the numerical simulation direct shear test and laboratory direct shear test under four different vertical pressures are consistent. See Table 4 for the final calibration results.

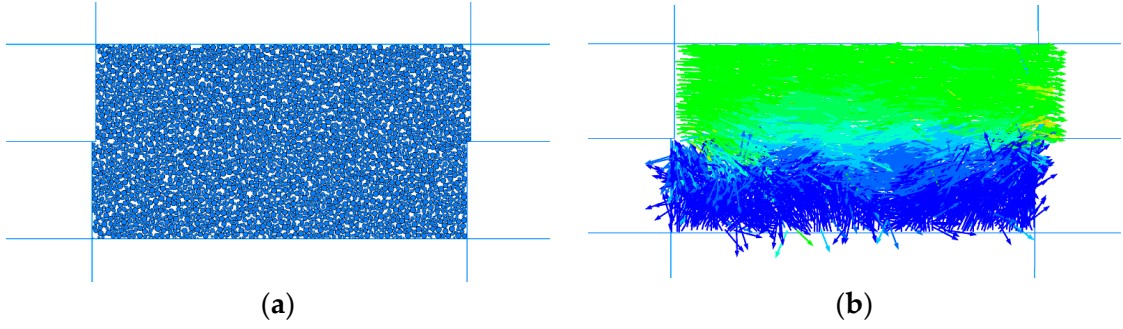

(**a**)　　　　　　　　　　　　　(**b**)

**Figure 6.** (**a**) Shear test. (**b**) Displacement vector diagram of shear test.

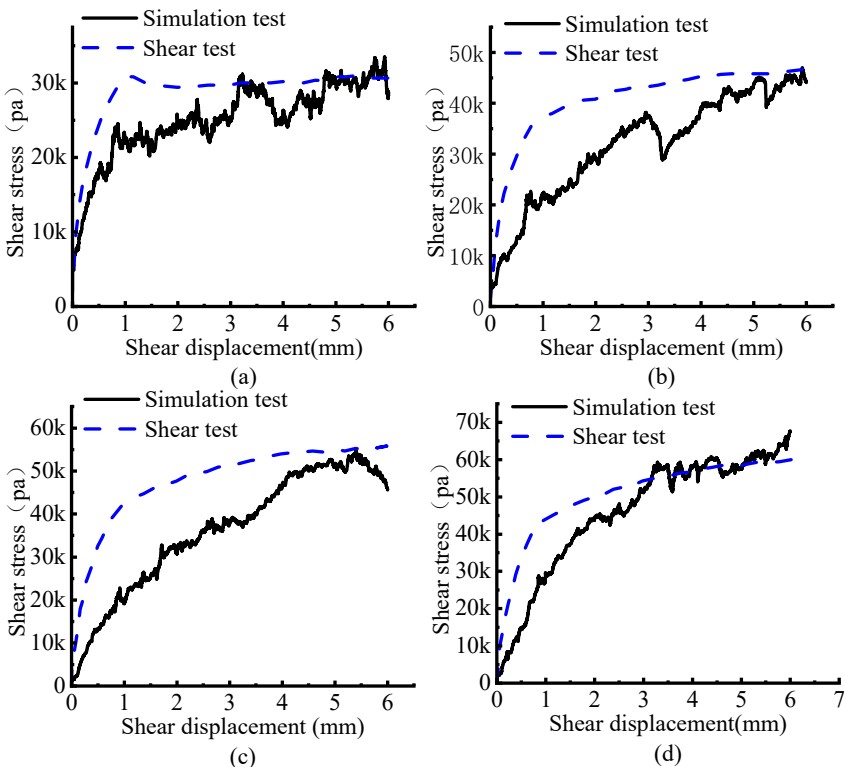

**Figure 7.** Shear stress–displacement curve. (**a**) Vertical pressure 60 kpa. (**b**) Vertical pressure 120 kpa. (**c**) Vertical pressure 180 kpa. (**d**) Vertical pressure 240 kpa.

**Table 4.** Microparameters of soil particles.

| Effective Modulus | Stiffness Ratio | Tensile Strength (pa) | Shear Strength (pa) | Friction Coefficient |
|---|---|---|---|---|
| $7.8 \times 10^7$ | 3 | $2.3 \times 10^6$ | $1.3 \times 10^6$ | 25 |

## 3. Results and Discussion

Using PFC-2D software, the aggregate models of the arrester bed with soil content of 0%, 5%, 10%, 15%, 20%, 25%, 30% and 35% were established, respectively. The wheel load was set as 2.47 t and the driving speed was set as 50.80 km/h (14.11 m/s). The numerical simulation test recorded the uncontrolled vehicle's stopping distance and wheel subsidence depth under the same load and driving speed.

During the emergency stop of the vehicle, the main energy consumption modes of vehicle kinetic energy include dynamic energy conversion between wheels and aggregate particles, compaction resistance, bulldozing resistance, air resistance and ramp resistance. Except for air resistance and ramp resistance, these energy consumption modes are all related to the amount of wheel subsidence. The resistance to the wheel is shown in Figure 8.

Aggregate pollution will affect the flow of aggregate gravel particles, reducing wheel subsidence and ultimately increasing the stopping distance of uncontrolled vehicles. There are two factors affecting the wheel subsidence depth. On the one hand, it comes from vehicle factors such as load and wheel size, and on the other hand, it comes from road factors. When the aggregate flow in the truck escape ramps is strong, the wheel is more likely to fall into aggregate under the same load. The wheel and aggregate have a greater contact angle $\theta$, which makes the resistance $F_x$ in the horizontal direction of the vehicle larger and the stopping distance shorter. However, with the increase in aggregate pollution, the fluidity of aggregate will change.

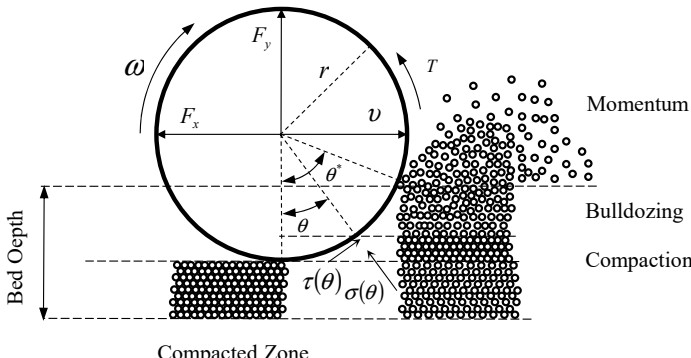

**Figure 8.** Interaction between wheels and aggregates.

The force on the wheel in the horizontal direction can be described as:

$$F_x = rb \left\{ \int_\theta^{\theta^*} \tau(\theta) \cos\theta d\theta - \int_\theta^{\theta^*} \sigma(\theta) \sin\theta d\theta \right\} \tag{3}$$

The force on the wheel in the vertical direction can be described as:

$$F_y = rb \left\{ \int_\theta^{\theta^*} \tau(\theta) \sin\theta d\theta + \int_\theta^{\theta^*} \sigma(\theta) \cos\theta d\theta \right\} \tag{4}$$

The external torque on the wheel can be described as:

$$T = r^2 b \left\{ \int_\theta^{\theta^*} \tau(\theta) d(\theta) \right\} \tag{5}$$

When the soil content in the contaminated aggregate is 0–15%, as shown in Figure 9, the soil particles mainly exist in the gap with the aggregate skeleton. In this state, the soil content in the contaminated aggregate is relatively small, and the mobility limitations of the aggregate are limited. The internal contact state of the aggregate is mainly dominated by the rigid contact between the gravel particles, and the macroscopic mechanical performance of the aggregate is similar to that of the nonpolluting aggregate. When the aggregate in this state is subjected to the pressure of the incoming wheel, it can destroy the bond contact between a small amount of soil particles under pressure and shear force, and only a small amount of soil particles can show the bond characteristics. Moreover, compared with the nonpolluting aggregate, the polluted aggregate particles will increase the aggregate's positive stress and shear stress to the wheel to increase the resistance in the horizontal direction. In contrast, the fluidity changes and the wheel subsidence is reduced. As shown in Figure 10, when the soil content in the contaminated aggregate is 5%, the stopping distance of the uncontrolled vehicles on the aggregate in this state is slightly shorter in the case of little change in subsidence. In summary, when the soil content in the contaminated aggregate is 0–15%, the aggregate is in good condition. Compared with the stopping distance of the nonpolluting aggregate, the change in the stopping distance is less than 5%. It is not necessary to maintain the aggregate, and its performance is consistent with that of the nonpolluting aggregate.

When the soil content in the contaminated aggregate reaches 20%, as shown in Figure 11, with the increase in soil particles, the soil particles exist between the skeletons of aggregate particles and the increase from the upper to the bottom of the aggregate. In this state, the cohesive contact in the internal contact of the aggregate increases, and the macroscopic mechanical manifestation of the aggregate begins to be affected by the soil's cohesive contact and the gravel aggregate's rigid contact. The fluidity of the aggregate begins to decrease significantly, and the wheel subsidence is significantly reduced. The contact angle between the wheel and the particle aggregate is also reduced. Although the wheel's normal stress and shear stress are also increased, the decrease in the contact angle

between the wheel and the particle aggregate significantly impacts the resistance in the horizontal direction. As shown in Figure 12, the stopping distance of the uncontrolled vehicles began to increase significantly from this stage. When the soil content in the contaminated aggregate reached 20–25%, compared with the nonpolluting aggregate, the stopping distance of the uncontrolled vehicles increased by more than 20%. In order to prevent further change in the aggregate state, it is necessary to carry out the loose refurbishment of aggregate in time to avoid the continuous degradation of the performance of the truck escape ramps.

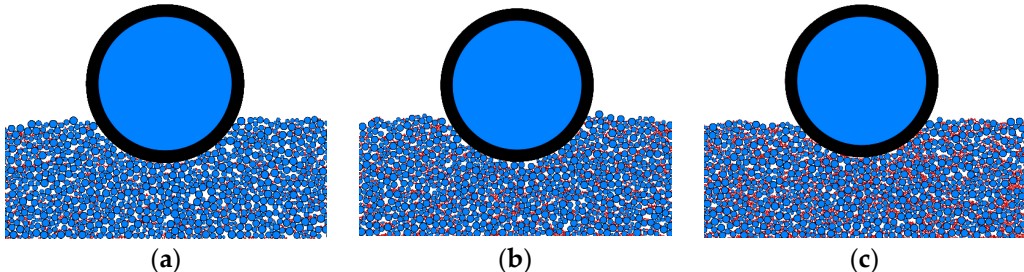

**Figure 9.** Subsidence of wheels on contaminated aggregates. (**a**) Soil content in aggregate 5%; (**b**) Soil content in aggregate 10%; (**c**) Soil content in aggregate 15%.

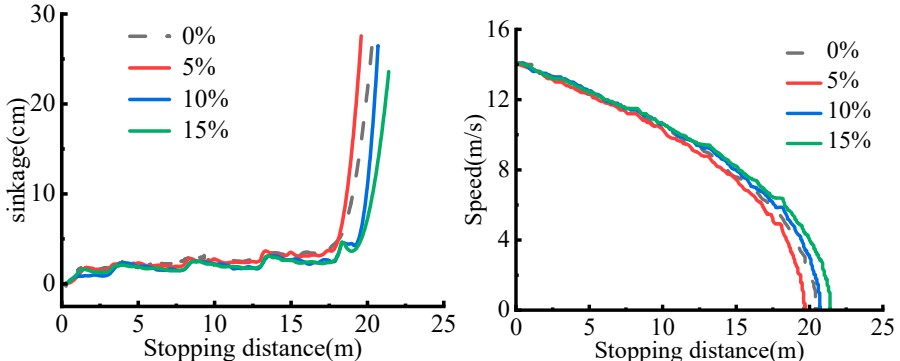

**Figure 10.** Stopping distance of uncontrolled vehicles on contaminated aggregate.

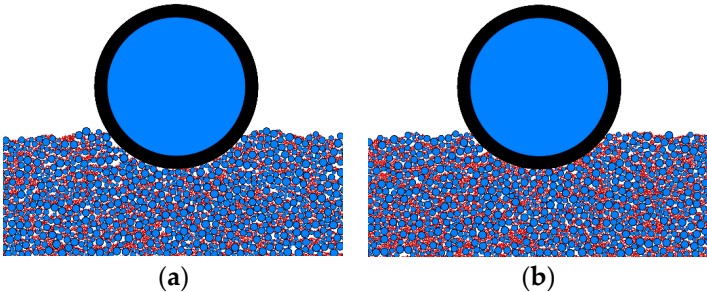

**Figure 11.** Subsidence of wheels on contaminated aggregates. (**a**) Soil content in aggregate 20%; (**b**) Soil content in aggregate 25%.

As shown in Figure 13, when the soil content in the contaminated aggregate reaches 30–35%, the soil particles are filled between the aggregate particle skeletons. The influence of cohesive contact of soil particles on the aggregate is further enhanced. With the increase in cohesive contact, the aggregate's overall shear and compression resistance are enhanced, and the subsidence of the wheel is further reduced. As shown in Figure 14, the stopping distance increases by more than 50%, which makes the original reasonable truck escape ramps fail to meet the design requirements, resulting in the vicious accident of the uncontrolled vehicles rushing out of the truck escape ramps. At this time, the soil content in the aggregate is too much. Even if the aggregate is maintained by loose and refurbishment

methods, the fluidity will decrease significantly again in a short time under the influence of rainwater and the gravity of the aggregate particles. It is suggested that the aggregate at this time be sieved or replaced.

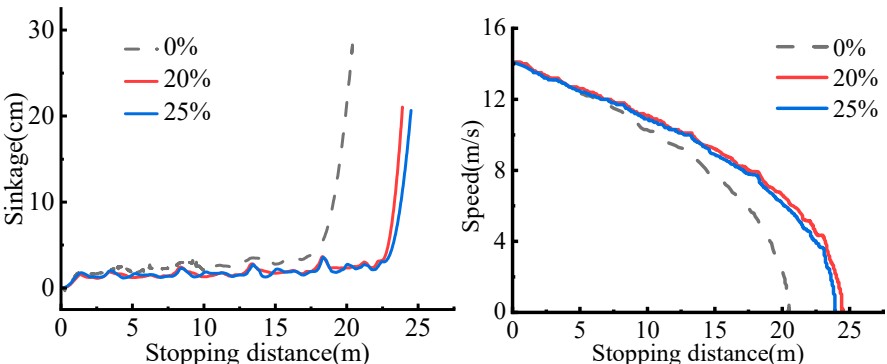

**Figure 12.** Stopping distance of uncontrolled vehicles on contaminated aggregate.

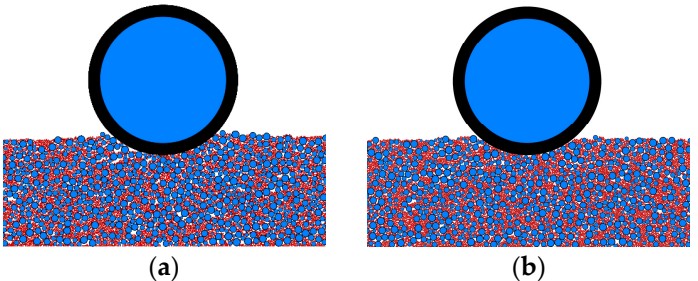

**Figure 13.** Subsidence of wheels on contaminated aggregates. (**a**) Soil content in aggregate 30% (**b**) Soil content in aggregate 35%.

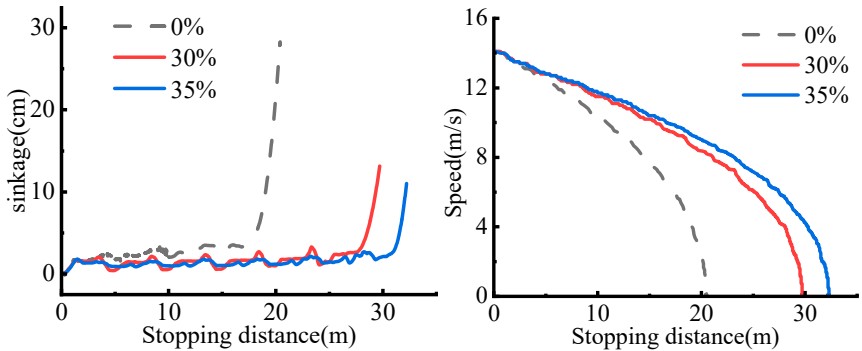

**Figure 14.** Subsidence of wheels on contaminated aggregates.

## 4. Conclusions

Aggregate pollution will affect the fluidity of aggregate gravel particles. With the increase in soil content in the polluted aggregate, the fluidity of the aggregate will worsen, and the subsidence of the wheel will decrease. The decrease in the wheel subsidence results in a decrease in the force of the aggregate on the wheels in the horizontal direction, increasing the stopping distance of the uncontrolled vehicles.

With the increase in aggregate pollution, the stopping distance increases correspondingly. When the soil content in the aggregate is 0–15%, the aggregate is in good condition, and the increased range of the stopping distance of the uncontrol vehicle is within 5%. At this time, it is recommended that the aggregate of the arrester bed should not be maintained; when the soil content in the aggregate exceeds 20%, the stopping distance increases by 20%. At this time, it is recommended that the arrester bed be loosely renovated and maintained; when the soil content in the polluted aggregate exceeds 30%, the increase in stopping

distance will reach more than 50%. As the gaps between aggregates in this state are full of soil, replacing the aggregates or screening them for soil removal is recommended.

In this paper, the influence of aggregates with different degrees of pollution on stopping distance was studied by taking silty cohesive soil as a typical pollutant. The influence of other soil-contaminated aggregates on stopping distance can be further studied. In addition to soil pollution, the impact of fuel leakage pollution, cargo-scattering pollution, and aggregate gravel corrosion degradation on vehicle stopping distance can also be further studied in the future, so as to realize the sustainable development of truck escape ramps in the construction of road safety facilities.

**Author Contributions:** Conceptualization, P.Q.; methodology, P.Q.; software, Z.L.; validation, Z.L.; formal analysis, Z.L.; investigation, H.L. and G.W.; resources, P.Q.; data curation, Z.L.; writing—original draft preparation, P.Q., Z.L. and H.L.; writing—review and editing, Z.L. and H.L.; visualization, J.H. and G.W.; supervision, J.H.; project administration, P.Q.; funding acquisition, J.H. All authors have read and agreed to the published version of the manuscript.

**Funding:** The authors disclosed receipt of the following financial support for the research, authorship, and publication of this article: Guangxi Science and Technology Major Project(grant number: GuikeAA22068061; GuikeAA22068060) and Guangxi Natural Science Foundation Project(grant number: 2019JJA160121).

**Acknowledgments:** Thanks to Qin Pinpin and the members of the research group for their help. Without their guidance and help, the research could not have been completed.

**Conflicts of Interest:** The authors declare no conflict of interest.

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
