# Peer review of "Influence of Aggregate Pollution in Truck Escape Ramps on Stopping Distance of Uncontrolled Vehicles"

_sustainability, doi:10.3390/su141811593_

Round 1

Reviewer 1 Report

Let me preface this by saying that I am not a mechanical engineer, but a civil engineer specialising in transport planning, so I focused more on the general structure of the article than on the details. As far as I am concerned, I think you should improve the readability of your paper. I enclose a PDF with many notes and suggestions. The notes are all in anonymous form.

Author Response

Response to Reviewer 1 Comments

Dear Reviewer:

Thank you for your comments on our manuscript entitled "Influence of Aggregate Pollution in Truck Escape Ramps on Parking Distance of Uncontrolled Vehicles" (ID: sustainability-1807007). Those comments are all valuable and very helpful for revising and improving our paper, as well as the important guiding significance to our researches. We have studied comments carefully and have made correction which we hope meet with approval. Revised portion are marked in red in the paper. The main corrections in the paper and the responds to the reviewer 's comments are as follows:

Point 1: What does this mean? That the procedure involves the simulation of 5 m long section of aggregate? Maybe it is better to clarify.

Response 1: Thank you for this question. The reason we divide aggregate into 5m segments is to reduce the amount of calculation. Before the numerical simulation, we carried out tests on our computer. With the calculation capacity of our computer, when the aggregate is divided into 5m sections, the calculation efficiency is the highest. We have modified the paper according to your suggestions and added the corresponding description.

Point 2: 2.It is probably my fault, but I do not understand the meaning of 'balanced' in this context.

Response 2: Thank you for this question. This is our problem in writing the paper, for which we are sorry. In this paper, ” balanced “ means the balance of interaction forces between particles.

Point 3: I expect a peak stress increasing with the increase of the relative density, and this is confirmed by the real data. But the simulation has lower values at 45% of relative density. Please justify.

Response 3: Thank you for this question. Because there is a critical point of friction and interlocking action between particles. When the pressure loaded in the tests exceeds the critical point, the peak pressure will change abruptly and then fall back. The critical point is low when the particles are loose, but we have not studied how much. Although we cannot accurately determine the critical point, the peak pressure will change abruptly within a certain range of the critical point. In the numerical simulation test, the peak pressure of the test group with small relative density is slightly higher than that of the test group with large relative density, but the average stress of the test group with small relative density is smaller than that of the test group with large relative density. Since the test data comes from the Al-Qad test in Reference ” Use of Gravel Properties to Develop Arrester Bed Stopping Model “. There is only peak stress in the test data, so we only compare the peak stress in the paper.

According to your other opinions, we have revised the paper in detail. Please refer to the part marked with red in the paper for details .

  1. We modified the introduction.
  2. In lines 111-117 of Chapter 2, we added the introduction and explanation of ”PFC “ software, and modified the unclear expression.
  3. In lines 147-150 of Section 2.1, we modified the punctuation according to the reviewer 's comments.
  4. In lines 160-161 of Section 2.2, we have modified the format of the references.
  5. In lines 163-164 of Section 2.2, we add an explanation of why we chose to segment the aggregate as 5m.
  6. In lines 167-168 of Section 2.2, we elaborate more specifically on “ balanced “ based on the questions posed by the reviewers.
  7. In lines 167-168 of Section 2.2, we explained the ” balanced “ in more detail according to the questions raised by the reviewers.
  8. In lines 194-198 of Section 2.4, we added a detailed description of the calculation method of stopping distance according to the opinions of the reviewers.
  9. In lines 210-211 of Section 2.4, we corrected sentences with ambiguous expressions.
  10. In line 213 of Section 2.4, we corrected ”parking distance“ to ”stopping distance“.
  11. In line 231 of Section 2.5, we corrected the wrong unit format.
  12. In lines 238-241 of section 2.5, we added the illustration according to the opinions of the reviewers.
  13. In lines 249-256 of Section 3, we have modified the sentences with poor language expression.
  14. In line 318 of Section 3 , we corrected ” parking distance “ to ” stopping distance “.

We appreciate for Editors and Reviewers' warm work earnestly, and hope that the correction will meet with approval. Once again, thank you very much for your comments and suggestions.

Reviewer 2 Report

1. How the numerical simulations can be assumed to be representing a real world phenomena is not clear.

2. The stopping sight distance calculations are not clear and how the basic objective of  measuring parking distance is not clear.

3. Stooping distance is more influenced by speed than any other parameters. Then what is the purpose of doing this study?

4. There is more of a stress on numerical simulation of soil properties rather than on the effect on stopping sight distance.

Author Response

Response to Reviewer 2 Comments

Dear Reviewer:

Thank you for your comments on our manuscript entitled "Influence of Aggregate Pollution in Truck Escape Ramps on Parking Distance of Uncontrolled Vehicles" (ID: sustainability-1807007). Those comments are all valuable and very helpful for revising and improving our paper, as well as the important guiding significance to our researches. We have studied comments carefully and have made correction which we hope meet with approval. Revised portion are marked in red in the paper. The main corrections in the paper and the responds to the reviewer 's comments are as follows:

Point 1: How the numerical simulations can be assumed to be representing a real world phenomena is not clear.

Response 1: Thank you for this question. In order to make the numerical simulation consistent with the real situation, we have taken these measures : First of all, We choose the proven and reliable numerical simulation method. The discrete element method used in this paper has been widely used in the field of wheel-soil and truck escape ramps. It is proved that it is feasible to use the discrete element method to study the problems related to the truck escape ramps. Secondly, The paper calibrated the constitutive model of wheel-aggregate and soil. The model is calibrated by gravel triaxial compression test, soil direct shear test and real vehicle test to ensure the accuracy of the model. The paper focuses on the stopping distance. In the calibration of numerical simulation, the average maximum error between the numerical simulation stopping distance and the real vehicle test parking distance is less than 10 %, which is within the acceptable error range.

Point 2: The stopping sight distance calculations are not clear and how the basic objective of  measuring parking distance is not clear.

Response 2: Thank you for this question. The paper does have the problem of unclear description of the calculation method of parking sight distance. This is our negligence in writing the paper, for which we apologize. We have made corresponding improvements in the paper. We use the wheel-aggregate model to simulate the stopping process, so we simplify the vehicle as the wheel, and the calculation of the stopping distance is based on the wheel. The calculation method of stopping distance is as follows: the starting point is the position at which the wheel is given the initial driving speed, and the ending point is the position at which the wheel finally stops after consuming energy on the arrester bed of the truck escape ramps. The driving distance of the wheel on the way is the stopping distance of the vehicle.

Point 3: Stoping distance is more influenced by speed than any other parameters. Then what is the purpose of doing this study?

Response 3: Thank you for this question. There are two factors affecting the stopping distance of uncontrolled vehicle on the truck escape ramps. On the one hand is the vehicle factors, such as vehicle load and vehicle speed, on the other hand is the impact of truck escape ramps, such as the slope, length and aggregate material of the truck escape ramps. The paper focuses on the influencing factors of the truck escape ramps. The performance degradation of the truck escape ramps caused by the change of aggregate state is a common problem in engineering application. The purpose of this project is to study the influence of the change of the aggregate state of the truck escape ramps on the stopping distance, and to provide the basis for the design of the length of the arrester bed of the truck escape ramps and the subsequent maintenance.

Point 4: There is more of a stress on numerical simulation of soil properties rather than on the effect on stopping sight distance.

Response 4: Thank you for your suggestion. The soil properties in aggregates are very complex. In this paper, we only studied the influence of different soil content in aggregate on stopping distance. The influence of more complex soil properties on stopping distance will be our next research direction. The influence of more complex soil properties on stopping distance will be our future research direction.

we have revised the paper in detail. Please refer to the part marked with red in the paper for details.

  1. We modified the introduction.
  2. In lines 111-117 of Chapter 2, we added the introduction and explanation of “ PFC “ software, and modified the unclear expression.
  3. In lines 147-150 of Section 2.1, we modified the punctuation according to the reviewer 's comments.
  4. In lines 160-161 of Section 2.2, we have modified the format of the references.
  5. In lines 163-164 of Section 2.2, we add an explanation of why we chose to segment the aggregate as 5m.
  6. In lines 167-168 of Section 2.2, we elaborate more specifically on “ balanced ” based on the questions posed by the reviewers.
  7. In lines 167-168 of Section 2.2, we explained the "balanced" in more detail according to the questions raised by the reviewers.
  8. In lines 194-198 of Section 2.4, we added a detailed description of the calculation method of stopping distance according to the opinions of the reviewers.
  9. In lines 210-211 of Section 2.4, we corrected sentences with ambiguous expressions.
  10. In line 213 of Section 2.4, we corrected "parking distance" to "stopping distance".
  11. In line 231 of Section 2.5, we corrected the wrong unit format.
  12. In lines 238-241 of section 2.5, we added the illustration according to the opinions of the reviewers.
  13. In lines 249-256 of Section 3, we have modified the sentences with poor language expression.
  14. In line 318 of Section 3 , we corrected "parking distance" to "stopping distance".

We appreciate for Editors and Reviewers' warm work earnestly, and hope that the correction will meet with approval. Once again, thank you very much for your comments and suggestions.

Round 2

Reviewer 1 Report

Thank you for your corrections. You gave me a detailed answer for point 3, but I would suggest to give some indications also inside the text.

In general terms I have a positive evaluation of your article, but I think that you could still improve the fluidity of the text by avoiding too short sentences. These short sentences in my opinion give the impression of a fragmented text. However, I understand that this is a matter of text quality which has nothing to do with the scientific value of the article.

Author Response

Dear Reviewers:

Thank you for your comments on our manuscript entitled "Influence of Aggregate Pollution in Truck Escape Ramps on Parking Distance of Uncontrolled Vehicles" (ID: sustainability-1807007).  Those comments are all valuable and very helpful for revising and improving our paper, as well as the important guiding significance to our researches. We have studied comments carefully and have made correction which we hope meet with approval. Revised portion are marked in red in the paper.

According to your suggestion, we added a description of the test in lines 187-191 of the paper. We have made some modifications to the English expression of the paper. After the paper is accepted, we will choose to use the paper editing service recommended by the editor to further improve the smoothness of the paper.

We appreciate for Reviewers' warm work earnestly, and hope that the correction will meet with approval. Once again, thank you very much for your comments and suggestions.
